# Current clinical practice in managing somatosensory impairments and the use of technology in stroke rehabilitation

Ananda Sidarta[1]*, Yu Chin Lim[1], Russell A. Wong[2], Isaac O. Tan[1], Christopher Wee Keong Kuah[1,3], Wei Tech Ang[1,2,4]

**1** Rehabilitation Research Institute of Singapore, Nanyang Technological University, Singapore, Singapore, **2** Lee Kong Chian School of Medicine, Nanyang Technological University, Singapore, Singapore, **3** Centre for Advanced Rehabilitation Therapeutics (CART), Tan Tock Seng Hospital, Singapore, Singapore, **4** School of Mechanical & Aerospace Engineering, Nanyang Technological University, Singapore, Singapore

\* ananda.sidarta@ntu.edu.sg

**Data Availability Statement:** All relevant data are within the paper and its Supporting information files.

## Abstract

Stroke-induced somatosensory impairments seem to be clinically overlooked, despite their prevalence and influence on motor recovery post-stroke. Interest in technology has been gaining traction over the past few decades as a promising method to facilitate stroke rehabilitation. This questionnaire-based cross-sectional study aimed to identify current clinical practice and perspectives on the management of somatosensory impairments post-stroke and the use of technology in assessing outcome measures and providing intervention. Participants were 132 physiotherapists and occupational therapists currently working with stroke patients in public hospitals and rehabilitation centres in Singapore. It was found that the majority (64.4%) of the therapists spent no more than half of the time per week on somatosensory interventions. Functional or task-specific training was the primary form of intervention applied to retrain somatosensory functions in stroke survivors. Standardised assessments (43.2%) were used less frequently than non-standardised assessments (97.7%) in clinical practice, with the sensory subscale of the Fugl-Meyer Assessment being the most popular outcome measure, followed by the Nottingham Sensory Assessment. While the adoption of technology for assessment was relatively scarce, most therapists (87.1%) reported that they have integrated technology into intervention. There was a common agreement that proprioception is an essential component in stroke rehabilitation, and that robotic technology combined with conventional therapy is effective in enhancing stroke rehabilitation, particularly for retraining proprioception. Most therapists identified price, technology usability, and lack of available space as some of the biggest barriers to integrating robotic technology in stroke rehabilitation. Standardised assessments and interventions targeting somatosensory functions should be more clearly delineated in clinical guidelines. Although therapists were positive about technology-based rehabilitation, obstacles that make technology integration challenging ought to be addressed.

**Funding:** AS - RFP/19002, funded by the Rehabilitation Research Institute of Singapore. The funders had no role in study design, data collection and analysis, decision to publish, or preparation of the manuscript.

**Competing interests:** The authors have declared that no competing interests exist.

## Introduction

Stroke is one of the leading causes of death and disability worldwide, where its burden has risen sharply from 1990 to 2019 [1]. About 86 million stroke survivors required rehabilitation services globally in 2019, higher than the number in 1990 by 85% [2]. Advances in medicine and healthcare have increased the survival rate, yielding a high number of people requiring long-term rehabilitation. For stroke survivors, motor impairments are often accompanied by deficits in limb position and movement senses or *proprioception*, which are important for motor control. About half of the stroke survivors typically experience loss of bodily sensation (somatosensation) in one or more modalities such as touch, position senses, temperature, and pressure [3–6].

Deficits in upper limb somatosensation are associated with reduced hand use and poorer sensorimotor integration, resulting in a decreased quality in fine motor control, object manipulation, and grip force regulation [7–10]. A patient holding a hot cup may be unable to perceive pressure and temperature optimally, resulting in increased danger of scalding. For the lower limb, impaired proprioception and light touch sensation have been found to impact gait speed [11, 12], limit independence in activities of daily living and the ability to balance [5, 13]. Therefore, disturbances in somatosensation result in, not only learned non-use of the affected limb, but also reduced emotional well-being, quality of life, and compromised safety [4, 7, 14]. Remarkably, somatosensory impairments usually receive less attention compared to the motor deficits which are relatively easier to assess and observe [15–17]. While clinicians generally believe that somatosensory impairment gradually recovers spontaneously, there are stroke survivors who are left with some degree of such deficits in the chronic phase beyond 6 months post stroke onset [18, 19]. Conducting somatosensory assessments can be challenging clinically, as it involves different sensory modalities which require long test routines. Not surprisingly, some prior work in the United States and Australia reported that not all therapists perform the standardised assessments and somatosensory-related interventions in their clinical practice [20, 21].

New technology has advanced healthcare and influenced how stroke rehabilitation can be delivered [22–24] to provide consistent, objective and motivational feedback. Some examples include upper limb robotic systems [25, 26], lower limb and balance training systems [27, 28], and wearable sensors [29]. In particular, robot-assisted therapy has been popular to deliver a higher dose of upper limb training which has been shown to provide moderate benefits [30]. Game-based virtual and augmented reality systems with motion sensing technology have also gained traction in rehabilitation. Microsoft Kinect (e.g. in [31]) or tablet games and applications (e.g. in [32]) can be used to deliver life-like task-based exercises. Another recent development is the application of non-invasive brain stimulation to enhance brain plasticity and restore balance in the cortical excitability post-stroke [33]. Considering the adoption of technology in stroke rehabilitation, its application as an assessment tool to evaluate recovery progression and changes in performances following an intervention period becomes attractive. Current clinical assessment systems employ ordinal scales that are known to be subjective and often unreliable due to low sensitivity [34, 35]. In contrast, modern machines are capable of giving a more precise and objective evaluation of patient progress. Some studies, however, still suggest that the adoption of technology by the therapists for stroke rehabilitation is found to be lacking [36, 37].

In this study, we sought to understand the perspectives and opinions of therapists in Singapore concerning standard practices in managing somatosensory impairments. Here, 'to manage' was used to mean both assessment (of impairment) and intervention (e.g., training or therapy sessions, exercises). Results from this study would be beneficial to identify the gap

between the current practice and evidence-based recommendations. We also examined the adoption of different types of rehabilitation technology and how well it had been integrated in the clinical setting. Focus was given to the application of robotic systems for proprioceptive interventions. Lastly, the study also identified obstacles to implementing technology in clinical practice.

## Materials & methods

### Design

A questionnaire-based cross-sectional study was conducted with occupational therapists and physiotherapists working in stroke rehabilitation across various healthcare settings in Singapore. An online, self-administered anonymous questionnaire was carried out between August 2021 and February 2022. The reporting of this study adheres to established standards for reporting web-based surveys, the Checklist for Reporting Results of Internet E-Surveys (CHERRIES) [38]. The study was approved by the Institutional Review Board of Nanyang Technological University, Singapore (protocol number: IRB-2020-10-012).

### Materials

The questionnaire was developed by the research team and adapted from prior studies [20, 37] that aimed to determine somatosensory assessment and treatment used by the therapists for stroke survivors, and the type of technology employed in stroke rehabilitation programmes. All items within the questionnaire were evaluated and refined by an occupational therapist and a physiotherapist who were part of the research team. The final questionnaire distributed using the web-based application Microsoft Forms consisted of 21 question items divided into four sections. Of the 20 closed-ended questions, six allowed the participants the chance to provide further details if 'other' or 'unsure' option was selected. The response options for closed-ended questions varied from single- or multiple-select choice type to 5-point Likert (from *strongly disagree* to *strongly agree*) or frequency rating scale ranging from *not available* to *regularly* (defined as >5 times a week). One open-ended question was created for the participating therapists to elaborate their thoughts regarding the overall topic of the questionnaire.

The first section included demographic questions regarding therapists' length of experience in stroke rehabilitation, practice setting, and proportion of time spent with stroke survivors. The second section dealt with the management of somatosensory impairment in routine clinical practice. This included questions on the types of intervention and assessment frequently applied and the usage frequency of common somatosensory-related interventions. In this questionnaire, somatosensory intervention referred to any targeted forms of training or exercise that aim at improving somatosensation. For example, tactile (e.g., touch discrimination of textures) or object (e.g., recognition of solid objects) discrimination, training of proprioception (e.g., position sense, identifying direction of limb movements), thermal stimulation, compression therapy using pneumatic compression devices and garments, and electrical or magnetic stimulation including but not limited to TENS (transcutaneous electrical nerve stimulation) and rPMS (repetitive peripheral magnetic stimulation). As somatosensation appears to be essential for balance control, any balance-related exercises were considered for inclusion. Repetitive practice of active movements, and functional training that involves sensorimotor integration such as fine motor control and postural adjustment were also regarded as being relevant.

Ratings of the perceived clinical importance of proprioception in stroke rehabilitation were covered in the third section. The last section explored therapists' views and experiences of using technology in stroke rehabilitation, in particular for retraining proprioception. Here,

'rehabilitation technology' was defined as innovations in machines or devices which help to maximise functions and reduce impairments, and at the same time, provide objective assessment in stroke rehabilitation. The predefined lists of equipment and technology options in this section were determined based on their common appearance in the rehabilitation research and to cover a broad set of possible options. A question on the main barriers and obstacles to incorporating technology into rehabilitation practice was also asked in the latter part of the section.

## Recruitment

Licensed occupational therapists and physiotherapists, who were actively involved in working with stroke survivors in Singapore, with at least 1-year of clinical experience were recruited by purposive sampling. In total, 15 public hospitals and rehabilitation centres in Singapore were approached for this survey. All participating therapists were enrolled through contact with the Head of Department, clinical supervisors, or senior therapists, where these points-of-contact served as an interface between the research team and the eligible participants. There was no direct contact between the researchers and therapists to ensure anonymity.

## Procedure

The points-of-contact were initially provided with a verbal outline of the study. Once they expressed interest and agreed to assist the research team in recruiting the occupational therapists and physiotherapists in their department, a brief recruitment message with study background, inclusion criteria, and a web link (URL) to the questionnaire was sent to them via emails or WhatsApp text messages. The message was then circulated within the respective department to invite those who fulfilled the inclusion criteria to take part. The therapists were allowed to complete the questionnaire in their own time and were informed that all questions were optional, and their participation was completely voluntary. Consent for participation was obtained online via the same web link prior to starting the questionnaire. Gift vouchers prepared by the research team as incentives for participation were handed out by points-of-contact to the therapists upon completion of the questionnaire.

## Data analysis

The questionnaire responses were extracted and input into SPSS statistical software, version 28 (IBM Corp., Armonk, N.Y., USA) for coding and analysis. All nominal and ordinal data were analysed using frequency. The response percentage did not always sum to 100% as participants were allowed to select multiple answers and skip questions which they were not comfortable answering (the overall percentage of missing values was 1.1%). All open-ended responses were reviewed and independently coded into predefined categories by two coders from the research team. Any discrepancies in the coding were resolved through discussion. Coded responses to 'other' items were included in the frequency analysis as mentioned above. Therapists' verbatim comments that were deemed relevant and useful to explain or support the results were presented in the main text, together with their case number as denoted by 'PN. xx' in brackets. Questionnaire responses were stored in a secured harddrive that could only be accessed by the principal investigator and the research team who performed the analysis.

## Results

A total of 132 participants from 13 healthcare sectors completed the questionnaire. On average, it took 18.34 minutes for them to go through all questions, and none terminated the

questionnaire early. The participants were between the ages of 23 and 54 (M = 32.38, SD = 5.93; 109 females), comprising 56.1% (*n* = 74) physiotherapists and 43.9% (*n* = 58) occupational therapists. The average work experience in stroke rehabilitation was 6.58 years (*SD* = 5.21). A Mann-Whitney U test indicated that no differences were observed in age (*U* = 1849, *p* = .293) and years of experience (*U* = 1929.50, *p* = .404) between physiotherapists and occupational therapists. Many of them (45.5%, *n* = 60) work in acute or restructured hospitals, followed by day rehabilitation centres (28%, *n* = 37), community hospitals (12.9%, *n* = 17), and nursing homes (8.3%, *n* = 11). About half (50.8%, *n* = 67) of the therapists work in an inpatient care setting, and most (35.6%, *n* = 47) provided rehabilitation services to patients with subacute and chronic stroke. The complete demographic characteristics of the participating therapists are illustrated in Table 1.

**Table 1. Participants demographic characteristics.**

| Characteristic | *n* | % | *M* | *SD* |
|---|---|---|---|---|
| Age | | | 32.38 | 5.93 |
| Years of experience in stroke care | | | 6.58 | 5.21 |
| Gender | | | | |
| Female | 109 | 82.6 | | |
| Male | 23 | 17.4 | | |
| Profession | | | | |
| Occupational therapist | 58 | 43.9 | | |
| Physiotherapist | 74 | 56.1 | | |
| Practice setting | | | | |
| Inpatient | 67 | 50.8 | | |
| Outpatient | 63 | 47.7 | | |
| Health facility | | | | |
| Acute or restructured hospital | 60 | 45.5 | | |
| Community hospital | 17 | 12.9 | | |
| Day rehabilitation centre | 37 | 28.0 | | |
| Nursing home | 11 | 8.3 | | |
| Other | 6 | 4.5 | | |
| Regional healthcare cluster | | | | |
| Central | 46 | 34.8 | | |
| West | 31 | 23.5 | | |
| East | 20 | 15.2 | | |
| Multiple, nationwide | 35 | 26.5 | | |
| Client type | | | | |
| Acute | 17 | 12.9 | | |
| Subacute | 28 | 21.2 | | |
| Chronic | 19 | 14.4 | | |
| Acute and subacute | 11 | 8.3 | | |
| Subacute and chronic | 47 | 35.6 | | |
| All types | 9 | 6.8 | | |
| Stroke care service time | | | | |
| <25% | 41 | 31.1 | | |
| 26–50% | 35 | 26.5 | | |
| 51–75% | 29 | 22.0 | | |
| >75% | 27 | 20.5 | | |

## Management of somatosensory impairment

Various types of training or interventions typically performed in the clinics to improve somatosensory functions in stroke survivors can be seen in Fig 1. Functional or task-specific training (99.2%, $n = 131$) was shown to be the most popular form of exercise, followed by movement-based exercises (89.4%, $n = 118$), and balance-related training (79.5%, $n = 105$). Regardless of the stroke phase, these were the three most common interventions delivered across patients to retrain their somatosensory functions. The therapists also reported the use of somatosensory-focused approaches: proprioception (59.8%, $n = 79$), object discrimination or recognition (41.7%, $n = 55$), tactile-based exercises (40.2%, $n = 53$), use of compression (14.4%, $n = 19$), and thermal stimulation (12.9%, $n = 17$). Over half (54.5%, $n = 72$) also indicated employing electrical or magnetic stimulation. One therapist who selected 'other' option reported the use of splinting as another method of intervention.

Approximately two-thirds of the therapists (64.4%, $n = 85$) spent half of their time or less providing somatosensory interventions to stroke survivors with somatosensory impairment, and only 34.1% ($n = 45$) spent more than half of their time per week. Two were unsure of how much time they spent on such training every week. When asked to report the usage frequency of certain intervention approaches earlier mentioned, the majority (72.7%, $n = 96$) of them reported conducting functional training regularly (more than five times a week), 18.2% ($n = 24$) at least two times per week, and 6.1% ($n = 8$) rarely (less than two times per week). With regards to electrical or magnetic stimulation, only 9.8% ($n = 13$) implemented it regularly, 28.8% ($n = 38$) sometimes, and 45.5% ($n = 60$) rarely. Further, a small proportion of the therapists (16.7%, $n = 22$) stated that they delivered proprioceptive training regularly, 43.2% ($n = 57$) sometimes, and 27.3% ($n = 36$) rarely. Similarly, tactile based exercises were applied regularly by only a few therapists (6.1%, $n = 8$), sometimes 23.5% ($n = 31$), and rarely 50.8% ($n = 67$).

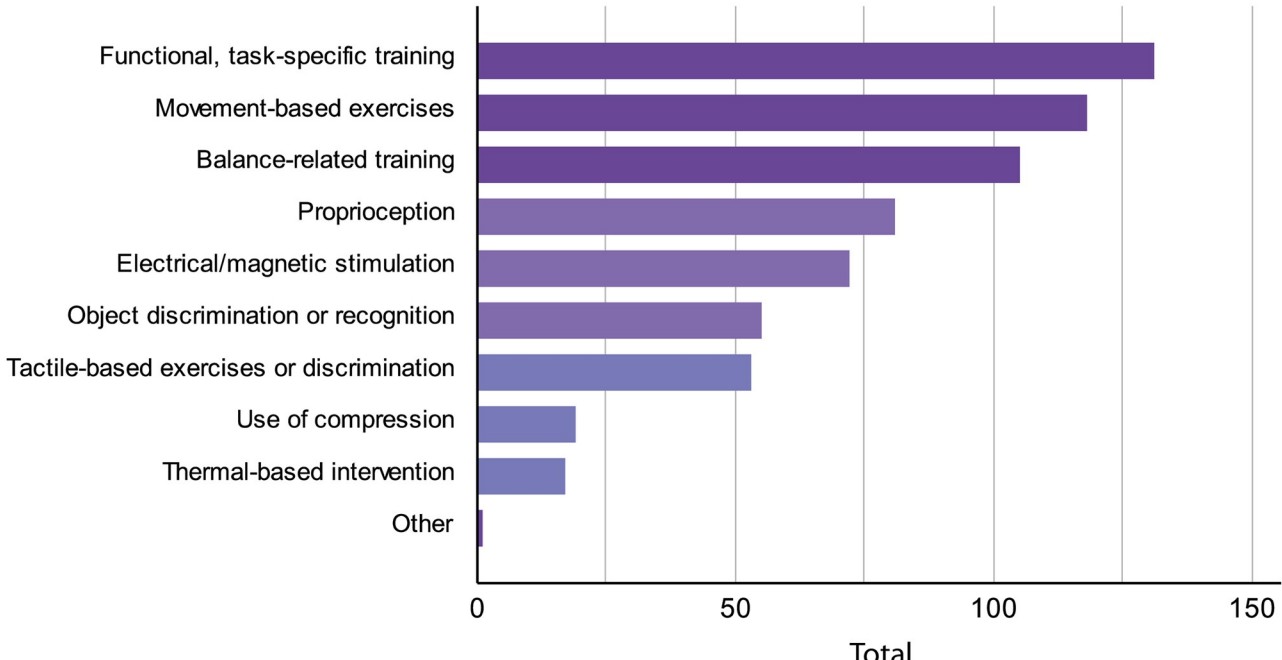

**Fig 1. Different forms of somatosensory-related intervention used in clinics.** The number of responses did not always sum to 132 as participants were allowed to select multiple answers.

**Table 2. Types of standardised and non-standardised assessment of somatosensation.**

|  | *n* | % |
|---|---|---|
| Standardised |  |  |
| Fugl-Meyer Assessment of Sensation | 43 | 32.6 |
| Nottingham Sensory Assessment | 7 | 5.3 |
| Rivermead Assessment of Somatosensory Performance | 5 | 3.8 |
| Semmes-Weinstein monofilament test | 3 | 2.3 |
| Non-standardised test | 129 | 97.7 |
| Non-standardised |  |  |
| Light touch | 94 | 71.2 |
| Position sense | 86 | 65.2 |
| Pain | 51 | 38.6 |
| Pressure | 45 | 34.1 |
| Sensory extinction | 43 | 32.6 |
| Stereognosis | 34 | 25.8 |
| Other | 3 | 2.3 |

Note. Participants were allowed to select multiple answers.

In terms of the standard clinical evaluation for somatosensation, the sensory scale of the Fugl-Meyer Assessment was employed by one-third of the therapists (32.6%, *n* = 43). Surprisingly, only a small minority employed the Nottingham Sensory Assessment (5.3%, *n* = 7), Rivermead Assessment of Somatosensory Performance (3.8%, *n* = 5), and Semmes Weinstein Monofilament Test (2.3%, *n* = 3). Generally, non-standardised outcome measures were preferred by the overwhelming majority of them (97.7%, *n* = 129), as opposed to the standardised assessment tools mentioned above. As illustrated in Table 2, the two most frequently used non-standardised tests were light touch (72.9%, *n* = 94) and position sense (66.7%, *n* = 86), followed by pain (39.5%, *n* = 51), pressure (34.9%, *n* = 45), sensory extinction (33.3%, *n* = 43), and stereognosis (26.3%, *n* = 34). In addition to the sensory modalities provided, three reported the evaluation of coordination, sharp-blunt discrimination, and thermal sensation through non-standardised measures.

### Technology adoption in stroke rehabilitation

The adoption of rehabilitation technology as part of regular clinical care is still fairly limited. Only about one-third (31.8%, *n* = 42) of the therapists had more than 3 years of experience in using rehabilitation technology, mainly in the inpatient setting (64.3%, *n* = 27) with patients in the subacute and chronic stroke phases (35.7%, *n* = 15). More than half (52.3%, *n* = 69) reported that they had fewer than 3 years of experience with rehabilitation technology, and about 16% (*n* = 21) had no experience (Fig 3(a)). Spearman's rank-order correlation found a moderate positive relationship between the therapists' years of experience in stroke care and with rehabilitation technology, ρ (129) = .53, *p* < .001, denoting that those who were more experienced usually had greater exposure to rehabilitation technology.

Although most therapists had little to no experience with rehabilitation technology, most still reported having access to certain forms of technology in their practice settings. Nearly all (95.5%, *n* = 126) stated that electrical stimulation devices were available to them, but less than half (48.4%, *n* = 61) used such devices two or more times per week and 51.6% (*n* = 65) rarely or never used them. The next most widely available forms of technology were virtual reality and commercial gaming systems (63.6%, *n* = 84), which were more accessible to those who

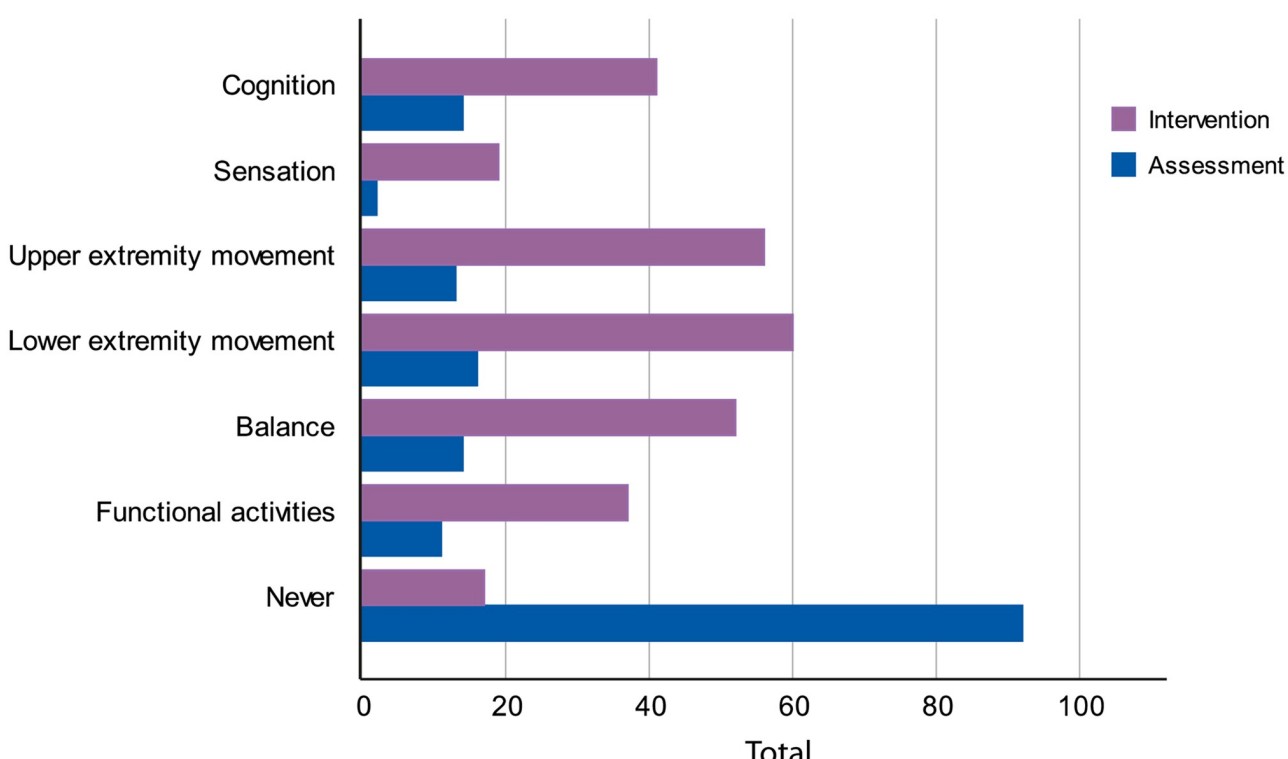

**Fig 2. Adoption of technology for the purpose of either assessment or intervention.** The number of responses did not always sum to 132 as participants were allowed to select multiple answers.

work in acute or restructured hospitals (50%, $n = 42$) and day rehabilitation centres (25%, $n = 21$). However, just over a quarter (26.2%, $n = 22$) of the therapists used them on a regular basis, and most (69%, $n = 58$) fewer than twice a week or never. Additionally, the proportion of therapists who stated having access to upper limb assistive technologies (47.7%, $n = 63$) was comparable to those who had access to lower limb assistive technologies (52.3%, $n = 69$). While upper limb assistive technologies were more accessible to those from acute or restructured (47.6%, $n = 30$) and community hospitals (22.2%, $n = 14$), assistive technologies for lower limb were more available to acute or restructured hospitals (55.1%, $n = 38$) and day rehabilitation centres (21.7%, $n = 15$). Both upper and lower limb assistive technologies were used at least two times a week by only 28.6% ($n = 18$) and 33.3% ($n = 23$) of the therapists, respectively. Few (5.3%, $n = 7$) reported having access to multicomponent technology that provides simultaneous training of different functioning abilities such as balance, cognition, and mobility.

Rehabilitation technology was employed primarily for intervention (87.1%, $n = 115$) compared to assessment (30.3%, $n = 40$). As presented in Fig 2, the most frequently reported technology-based intervention provided to stroke survivors was lower extremity movement (45.5%, $n = 60$), followed by upper extremity movement (42.4%, $n = 56$), balance (39.4%, $n = 52$), cognition (31.1%, $n = 41$), functional activities (28%, $n = 37$), and sensation (14.4%, $n = 19$). By contrast, technology was used by a relatively small proportion of the therapists to assess lower extremity movement (12.1%, $n = 16$), balance, cognition (both 10.6%, $n = 14$), upper extremity movement (9.8%, $n = 13$), functional activities (8.3%, $n = 11$), and sensation (1.5%, $n = 2$). Further analysis revealed that occupational therapists tended to incorporate technology in their practice to retrain upper limb and cognitive functions, whereas

(a) Years spent in using technology

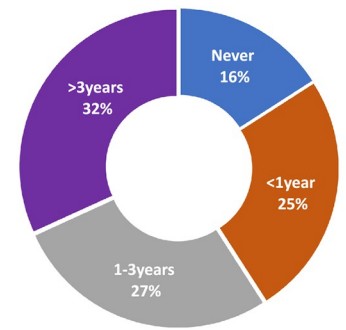

(b) Perceived barriers to integrating technology in clinical practice

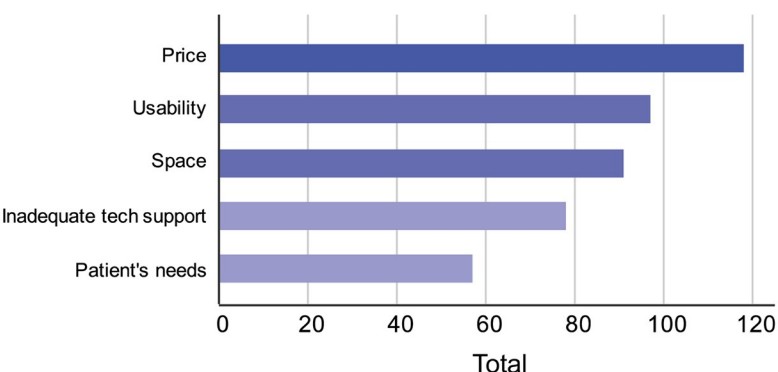

**Fig 3. Technology adoption in clinical practice.** (a) Therapists' years of experience with rehabilitation technology. (b) Perceived barriers to integrating technology in clinical practice. The number of responses did not always sum to 132 as participants were allowed to select multiple answers.

physiotherapists conducted technological intervention targeting lower limb functions and balance (see Fig F in S4 File).

## Perceived barriers to integrating robotic technology in clinical practice

A majority (89.4%, $n = 118$) of the participating therapists regarded price as the main barrier to the successful integration of robotic technology in clinical practice (see Fig 3(b)). Nearly all who work in the day rehabilitation centres (91.9%, $n = 34$), acute or restructured hospitals (90%, $n = 54$), and nursing homes (90.9%, $n = 10$) considered this the biggest implementation obstacle. Usability of technology or ease of use was identified by almost three-quarters (73.5%, $n = 97$) of the therapists as the next most common barrier. This was followed subsequently by lack of space (68.9%, $n = 91$), inadequate technical support (59.1%, $n = 78$), and patient's needs 43.2% ($n = 57$). Of particular note was most therapists (88.2%, $n = 15$) who work in the community hospitals ranked usability higher than price as the top barrier to implementation. Lack of available space was another challenge faced by 76.5% ($n = 13$) of those from community hospitals, which was of equivalent rank to price. On the other hand, inadequate technical support was one of the most common obstacles that prevent most therapists working in day rehabilitation centres (75.7%, $n = 28$) and nursing homes (63.6%, $n = 7$) from adopting robotic technology for rehabilitation. In addition to the predetermined list of options, some therapists mentioned that patient mobility, insufficient funding, and long setup time would affect the effective implementation of robotic technology.

Additional comparative analyses were conducted to examine therapists' views and practices on somatosensory assessment and intervention, as well as the adoption of technology in different regional healthcare clusters. Analyses in terms of speciality (physical therapist or occupational therapist) were also included. The full results can be seen in the "Supporting Information, S4 File".

## Perceived role of retraining proprioception and robot-assisted rehabilitation

A general positive evaluation of statements regarding the clinical importance of proprioceptive rehabilitation and the application of robotic technology in rehabilitation, particularly in proprioceptive retraining, is depicted in Fig 4. This is indicated clearly by considering the

### (a) Perceived clinical importance of position senses (proprioception) in stroke rehabilitation

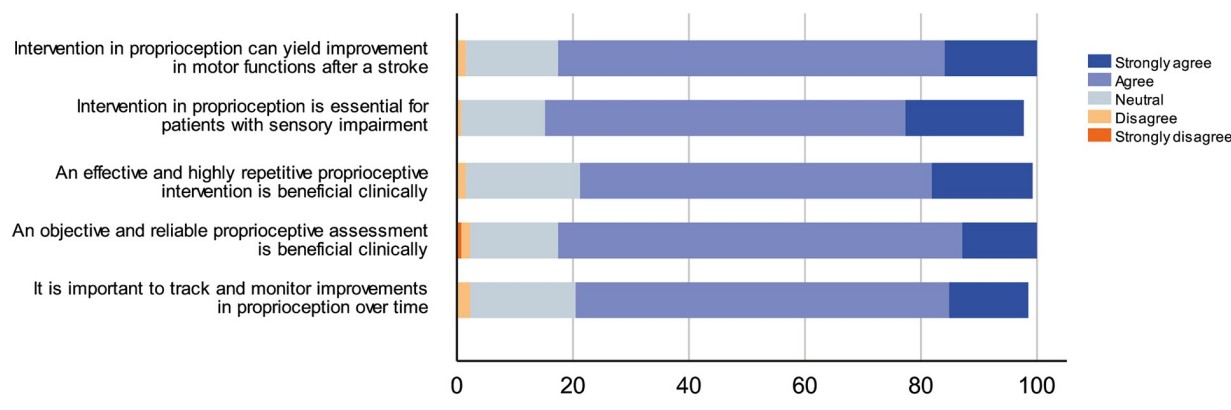

### (b) Perceived clinical importance of employing robotic technology in stroke rehabilitation

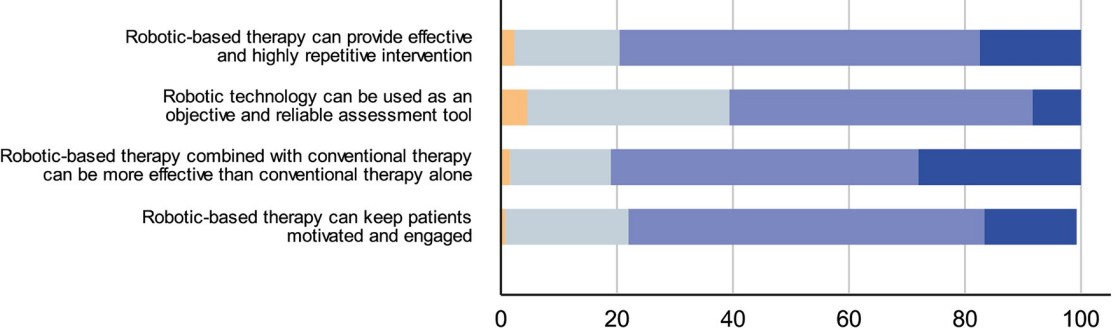

### (c) Perceived clinical importance of robotic technology in proprioceptive intervention

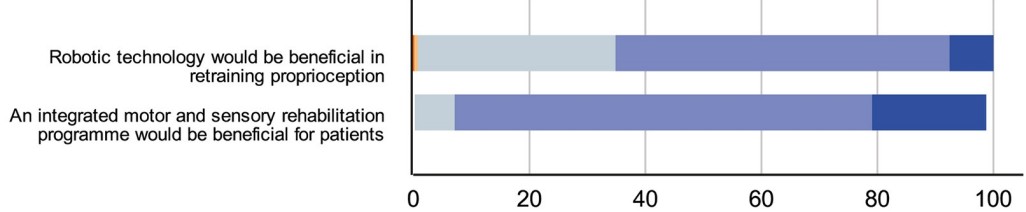

**Fig 4. Therapists' perspectives on the role of proprioception and technology adoption in stroke rehabilitation.** (a) Perceived clinical importance of *proprioception*. (b) Perceived clinical importance of robotic technology. (c) Perceived clinical importance of adopting robotic technology specifically for proprioception.

combined totals of the *agree* and *strongly agree* categories. A reliability analysis showed that the reliability and inter-item consistency were acceptable, $\alpha = 0.83$. The Cronbach's alpha of all items would not increase with the exclusion of any item, and thus no removal of items should be considered.

Fig 4(a) shows that most of the therapists (82.6%, $n = 109$) agreed that proprioceptive training can help to improve motor function following a stroke and is essential for patients with sensory impairment. This suggests that most therapists understood the benefits of retraining proprioception due to its impact on motor function, especially in stroke survivors with sensory loss. Further, 78% ($n = 103$) of them thought that an effective and highly repetitive intervention is beneficial to stroke patients, and there were more of them (82.6%, $n = 109$) appreciated the

idea of an objective and reliable assessment of proprioception. In addition, 78% ($n$ = 103) of therapists agreed that it is important to track and monitor the improvement of proprioception over time.

While most believed that an effective and highly repetitive intervention is clinically beneficial, 79.5% ($n$ = 105) of the therapists agreed that such an intervention can be achieved through robotic therapy (see Fig 4(b)). Although not as high as those who were in favour of robotic therapy, 60.6% of them ($n$ = 80) still considered that robotic-based assessments are objective and reliable. One physiotherapist seemed to appreciate technology-based rehabilitation due to its ability to lighten the workload of therapists, allowing them to focus on observing patients' performance and offering constructive feedback (PN. 2). By contrast, another physiotherapist thought that technology is unsuitable for patients with cognitive impairments and muscle weakness (PN. 88). Interestingly, 81% ($n$ = 70) of the therapists believed that robotic therapy combined with conventional therapy can be more effective than conventional therapy alone. This was echoed by one participant (PN. 20) who elaborated, "Technology needs to be used adjunct to conventional therapy and not in isolation to ensure carryover of skills," indicating that robotic technology is more likely to be adopted in interventions by combining it with conventional rehabilitation methods. This general level of agreement among the therapists further implies that most (77.3%, $n$ = 102) believed robot-assisted therapy can keep patients motivated and engaged. Despite this, when asked whether robotic technology would be beneficial for retraining proprioception, 65.2% ($n$ = 10) of them agreed (Fig 4(c)). One of them who did not share any opinion on this pointed out having a lack of knowledge of technology-based intervention options for somatosensory impairment (PN. 41). Moreover, therapists' opinion on employing rehabilitation programmes with integrated motor and sensory components was particularly strong, as 91.7% ($n$ = 107) of them believed that this would benefit the patients.

## Discussion

Stroke survivors require long-term rehabilitation to cope with their impairments and be independent in the community and home. People with stroke typically experience both motor and somatosensory impairments, but past literature suggests that somatosensory components receive lesser attention. Existing interventional studies that target somatosensory loss are still rather limited. One potential reason is the greater emphasis on movement-related interventions to regain functional independence, which are more widely studied and have observable benefits [17, 39, 40]. Another possible explanation is that therapists have inadvertently integrated somatosensory components into their daily rehabilitation routines, without targeting any specific sensory modalities. The findings in the second section of the questionnaire are in line with these arguments.

Therapists in the current study predominantly and regularly applied functional or task-specific training as the primary forms of somatosensory intervention. Movement-based exercises and balance training were reported to be the next most popular forms of intervention. Although these exercises simultaneously engage in sensory integration and postural control, they can be considered 'motor training'. And as highlighted by two participating therapists, motor improvements are more important than somatosensory aspects for functional independence (PN. 23, 97). On other hand, interventions focusing on sensations of touch (discerning roughness, pressure, or vibration) and temperature were less commonly implemented. This is interesting given that those types of sensory perception are arguably important for motor control and safety during movement. It is possible that stroke survivors with somatosensory deficits served by the participating therapists were of a small number. However, even if somatosensory intervention is required for some patients, most participants still spent less

than 25% of their clinical practice time providing targeted exercises. Two therapists who were unsure of their time spent on such intervention stated that other types of training intervention tend to take precedence over somatosensory interventions when somatosensory loss appears as a secondary impairment (PN. 97, 99). This is despite some evidence-based studies for more targeted interventions to retrain impaired tactile, proprioception, and other modalities [6, 39, 41, 42]. If present, such forms of targeted exercise are typically combined with functional or task-specific activities to effectively stimulate and strengthen post-stroke motor recovery in patients with motor and somatosensory deficits [43–46].

The current study found that the majority of therapists favoured non-standardised clinical measures for assessing somatosensory impairment. This corroborates another finding obtained in earlier surveys that examined the use of standardised somatosensory assessments in adult stroke survivors and children with neurological disorders [20, 21, 47]. These few studies revealed that the standardised assessments of somatosensation are underutilised. Evidence regarding the use of validated and reliable instruments by therapists in stroke rehabilitation varies across countries [48–51]. For example, the use of standardised assessments was notably high in the United Kingdom but relatively low in Canada, likely due to lack of time and knowledge about outcome measures. Although standardised outcome measures are commonplace in certain countries, international and local consensus regarding which instruments to use in practice is limited [49, 50]. This suggests that recommendations on the selection of valid and reliable outcome measures for stroke rehabilitation appear to be vague, hence non-standardised assessments were widely administered by therapists in the present study.

The two most common non-standardised assessments reported by the current participants are consistent with the results reported by Winward et al. [16], Doyle et al. [21], and Pumpa et al. [20], where light touch and proprioception were mostly assessed. Among the standardized clinical scales, the Fugl-Meyer Assessment having both motor and sensory subscales is widely used in practice to evaluate sensorimotor impairments post-stroke [52, 53]. This interesting finding is consistent with the idea that the Fugl-Meyer Assessment is considered holistic, and thus commonly adopted as part of the typical clinical assessments for the stroke population. While the Nottingham Sensory Assessment is a recommended and more detailed measure of various sensory deficits [54, 55], it was used less frequently than the sensory subscale of the Fugl-Meyer Assessment in this study. In fact, the original Nottingham Sensory Assessment with the application of specialist equipment is time-consuming and known to have poor inter-rater reliability; accordingly further revisions were made to improve the reliability and to reduce testing time [56]. Lastly, the Semmes-Weinstein Monofilament Test, which is found to be most frequently used by therapists in Australia [20], was the least popular among the provided standardised assessments.

The third section of the questionnaire explored participants' views towards proprioception, one of the specific somatosensory modalities which has been demonstrated to contribute greatly to motor control and learning. Unlike the work by Winward et al. that examined the clinicians' view on the importance of somatosensory assessment as a whole [16], we had a closer look at the interventional aspects of proprioception. Overall, participants in this study were positive about how proprioceptive intervention improves motor functions after stroke, particularly for those suffering from sensory impairment. This is consistent with recent training or exercise protocols which target proprioception and tactile senses [41, 57, 58]. In conjunction, the therapists also agreed strongly that an objective and reliable assessment would be useful to inform performance and track or monitor recovery progression. These could be achieved or facilitated by a highly repetitive rehabilitation programme with the use of robotic technology, which was agreeable to most therapists as shown in the final section of the questionnaire (Fig 4). Indeed, robot-assisted therapy has shown great promise, and recent studies

targeting proprioception have been proposed to address such dysfunction in the stroke population, e.g. as described in [59–61].

Most therapists demonstrated a strong preference for adopting rehabilitation technology for intervention. On the contrary, fewer agreed with the idea of technological-based assessment. This finding is in line with a recent study of American therapists being more likely to use technology for intervention than assessment [37]. Despite the recent development of robotic-based assessment systems [34, 62, 63], the acceptance of using these devices in clinical practice is still scarce, with one therapist explaining that assessment data generated are typically laborious to interpret, and thereby are impractical to inform clinical decision and progression (PN. 44). Such statements show that there is still much room for improvement in technology before there is widespread implementation. Another potential challenge is to develop a rehabilitation device that can evaluate the heterogeneous nature of somatosensory performance, including the discrimination and detection ability (PN. 54).

Rapid advances and innovation in robotic technology can address unmet challenges in rehabilitation in Singapore [23]. However, high price was perceived by the therapists to be the topmost barrier to integrating technology regardless of the practice setting. This finding echoed similar obstacles to integration identified in the previous study. For example, in a mixed methods survey, Li et al. identified perceived logistical issues (ease of use, storage space) and cost as the major factors in the United Kingdom [64]. Nonetheless, at least two studies suggest that robot-assisted rehabilitation does not significantly incur higher costs in the long run as compared to the conventional care [65, 66]. Therapists usually prefer technology that is easy to use, invokes less time to prepare, and compact (PN. 20, 44, 54, 113, 115). It would be worth noting that having a robotic system that can provide diverse assessment of somatosensory modalities makes more economical sense. Overcoming logistical issues would involve resource planning and cost-benefit analysis, which requires heavy discussion among stakeholders, such as healthcare managements and technology companies. This ensures that any robotic system developed can be translated into proper clinical use, such as in bedside testing or in telerehabilitation.

## Conclusions

This work determined the current clinical practice and perceptions in managing somatosensory impairment and implementing rehabilitation technology in stroke care within Singapore. Somatosensory-specific interventions and standardised assessments were rarely implemented in clinical practice. The current findings showed that technological applications in rehabilitation were more apparent in intervention than in assessment. Therapists believed that intensive training of proprioception and objective assessments are beneficial and critical to stroke recovery. Robotic technology can be seen as a way to promote standardisation in somatosensory assessments and to deliver more effective interventions. However, technology was rarely adopted for targeted sensory interventions by the participants. Lastly, price, ease of use, and space availability were viewed as the top three main obstacles to technology integration in clinical practice.

There are certain aspects of this study that warrant future investigation. Self-administered closed ended questions may limit responses and opinions and can lead to different interpretations of wording among different participants. Hence, methods such as focus group interviews with open ended questions can be employed as a follow-up study. Clinical guidelines that contain recommendations to promote more targeted training and standardised measures of somatosensory functions can also be developed. Technological innovation can change the rehabilitation landscape, but some real-world obstacles and barriers cannot be neglected. This

study was not designed to better understand customer needs or to increase the utilization rate of certain robotic technology. Therefore, more studies and evidence are vital to examine if the long-term benefits will outweigh the perceived obstacles.

## Supporting information

**S1 File. Questionnaire.**
(PDF)

**S2 File. CHERRIES checklist.**
(DOCX)

**S3 File. Coded data.**
(XLSX)

**S4 File. Additional analyses.**
(DOCX)

## Author Contributions

**Conceptualization:** Ananda Sidarta, Isaac O. Tan, Christopher Wee Keong Kuah, Wei Tech Ang.

**Data curation:** Ananda Sidarta.

**Formal analysis:** Yu Chin Lim, Russell A. Wong, Isaac O. Tan.

**Funding acquisition:** Ananda Sidarta, Wei Tech Ang.

**Methodology:** Ananda Sidarta, Yu Chin Lim, Christopher Wee Keong Kuah.

**Project administration:** Yu Chin Lim, Russell A. Wong.

**Supervision:** Ananda Sidarta, Christopher Wee Keong Kuah.

**Writing – original draft:** Ananda Sidarta, Yu Chin Lim, Russell A. Wong, Isaac O. Tan.

**Writing – review & editing:** Ananda Sidarta, Yu Chin Lim, Isaac O. Tan.

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
