## [Decision Letter · Decision Letter 0]

12 May 2022

PONE-D-22-10125Current Clinical Practices in Managing Somatosensory Impairments and the Use of Technology in Stroke RehabilitationPLOS ONE

Dear Dr. Sidarta,

Thank you for submitting your manuscript to PLOS ONE. After careful consideration, we feel that it has merit but does not fully meet PLOS ONE’s publication criteria as it currently stands. Therefore, we invite you to submit a revised version of the manuscript that addresses the points raised during the review process.

We look forward to receiving your revised manuscript.

Kind regards,

Benjamin A. Philip

Academic Editor

PLOS ONE

Journal Requirements:

Reviewers' comments:

Reviewer's Responses to Questions

**Comments to the Author**

1. Is the manuscript technically sound, and do the data support the conclusions?

Reviewer #1: Yes

Reviewer #2: Partly

2. Has the statistical analysis been performed appropriately and rigorously? 

Reviewer #1: Yes

Reviewer #2: Yes

3. Have the authors made all data underlying the findings in their manuscript fully available?

Reviewer #1: Yes

Reviewer #2: Yes

4. Is the manuscript presented in an intelligible fashion and written in standard English?

Reviewer #1: Yes

Reviewer #2: Yes

5. Review Comments to the Author

Reviewer #1: Detailing current clinical practice is an important endeavor and this article addresses key points that are often assumed but not empirically studied, such as the lack of clinical focus on sensory recovery, despite its importance in sensorimotor recovery post-stroke. Robotics-based interventions have been shown to result in moderate improvements post-training, though adoption seems to be slow and not match recommended guidelines. The authors aim to understand the adoption and use of technologies in stroke somatosensory rehabilitation using a questionnaire-based cross-sectional study of physiotherapists and occupational therapists in Singapore.

Intriguing, albeit not surprising, to learn that so few therapists utilize the more comprehensive somatosensory assessments, such as the Nottingham or Rivermead.

I had a difficult time finding fault in the author’s work. The nature of the questionnaire-based research is simple, though resulted in many interesting findings; the authors were meticulous about laying out all of these findings in a logical order that was easy to follow. They provide concrete goals for researchers interested in developing rehabilitation technologies with the goal of clinical adoption. The authors have created a well written research article full of findings that I feel will be useful to the rehabilitation research community. I look forward to citing the work myself.

Reviewer #2: Line 57-59 It is suggested to change the data to global data.

Line 62 Lack of supporting references. Besides, limb position sense is included in proprioception.

Line 93 There is lacking a bridge from the treatment component to the assessment component.

Line 105-106 The author mentioned that the gap between clinical practice and guide, but the paper did not mention the therapist practice as reference related guidelines.

Line 136-137 I think the area of the institution where the therapist works makes sense for research.As far as I know, Singapore is divided into many districts. Maybe a comparative analysis between different regions is worth studying.

Line 141 Why the author did not mention the “rehabilitation technology” here?

Line 143-144 Proprioception includes kinesthesia.

Line 154 Why the author focus on exploring therapists views and experiences of

using technology, especially proprioception.

163-170 How to ensure randomness and representation of sampling？How to ensure that the distribution of therapists' specialties matches the actual employment data in Singapore？The author mentions that participants are connected with superior therapists, but how to ensure that the participants connected with superior therapists are random and representative.

203-204 Since the author has collected the professional distribution of therapists, why not analyze their respective views on sensory assessment, sensory training, and new technical methods.I think the differences between the specialties of physical therapists and occupational therapists will provide different answers.

381-383 References are required.

396 I doubt that thermal sense is important for motion control rather than proprioception? Maybe there are other reasons to explain it.

435 The most commonly used assessment methods should be placed first and the least commonly used should be placed later.

6. PLOS authors have the option to publish the peer review history of their article (what does this mean?). If published, this will include your full peer review and any attached files.

Reviewer #1: **Yes: **Nathan A Baune

Reviewer #2: No

---

## [Author Response · Author response to Decision Letter 0]

8 Jun 2022

Dear Colleague,

We would like to thank the reviewers for their encouraging and constructive feedback with regards to our manuscript. We have carried out the modifications as suggested. Changes in the manuscript are shown in blue. Minor edits in the Abstract and minor typos in Figure 1 and 3 have been corrected. Updated supporting information (S3 and S4) has been included. A point-by-point response to the reviews is given below.

Reviewer #2: 

Line 57-59 It is suggested to change the data to global data.

As suggested, we have revised the first paragraph accordingly to present the global data, which reads the following: “Stroke has been known to be one of the world leading causes of death and disability, where its burden has risen sharply from 1990 to 2019 [1]. About 86 million stroke survivors required rehabilitation services globally in 2019, higher than the number in 1990 by 85% [2]” (Line 56-58)

Line 62 Lack of supporting references. Besides, limb position sense is included in proprioception. 

To avoid ambiguity, we only use ‘proprioception’ in the paragraph accordingly (Line 61). Supporting references to the prevalence of proprioceptive impairments can actually be found together with references written in the paragraph that follows (Line 64)

Line 93 There is lacking a bridge from the treatment component to the assessment component.

The bridge to link the treatment component to the assessment component has been added to illustrate the point raised by the reviewer (Line 93-98): 

“Considering the adoption of technology in stroke rehabilitation, its application as an assessment tool to evaluate recovery progression and changes in performances following an intervention period becomes attractive. Current clinical assessment systems employ ordinal scales that are known to be subjective and often unreliable due to low sensitivity [34, 35]. In contrast, modern machines are capable of giving a more precise and objective evaluation of patient progress.” 

Line 105-106 The author mentioned that the gap between clinical practice and guide, but the paper did not mention the therapist practice as reference related guidelines.

We have reworded the sentences to “Results from this study would be beneficial to identify the gap between the current practices and evidence-based recommendations”. Of which, we have illustrated some gaps between clinical practice of the therapists who responded in our survey, as well as some recent findings in the literature cited in the main text, e.g. in Winstein et al., 2016 and Schabrun et al., 2009. For example, we wrote that therapists chose other types of intervention over somatosensory-specific interventions despite prior recommendations for more targeted interventions to retrain impaired tactile, proprioception and other modalities (Line 407-411). We also mentioned that some therapists use proprioceptive interventions for stroke patients with sensory impairment as recommended by prior studies (Line 452-455). 

Line 136-137 I think the area of the institution where the therapist works makes sense for research. As far as I know, Singapore is divided into many districts. Maybe a comparative analysis between different regions is worth studying.

Indeed, the reviewer is correct. Healthcare services here are divided into 3 primary districts (or clusters), i.e. the west, central, and east. The fourth category includes nationwide services, having a few smaller day-care rehabilitation centres in multiple clusters. We ensured each primary cluster has been represented in our sampling. Table 1 has been updated accordingly. As suggested, we have also conducted the same sets of analyses and the results which are found in the Supporting Information S4 File. Briefly, the results of these sub-analysis are in consistent with the findings when taking the participants as a whole. For example, functional and task-specific training (including balance) were the most common types of somatosensory-related intervention in all healthcare clusters (Figure A in S4 File). 

Line 141 Why the author did not mention the “rehabilitation technology” here?

We wanted to define what constitutes somatosensory intervention without specifically talking about how to achieve or perform the intervention (e.g. using traditional tool versus recent technology). That is why we did not mention rehabilitation technology there. Indeed, we will only focus more on rehabilitation technology in the later section of the questionnaire.

Line 143-144 Proprioception includes kinesthesia.

We have reworded the paragraph accordingly. 

Line 154 Why the authors focus on exploring therapists views and experiences of using technology, especially proprioception.

We are focusing on proprioception, as it is important for motor control. In addition, modern rehabilitation technology has become more popular recently especially for upper-limb rehabilitation (e.g. Findlater and Dukelow, 2017). In conjunction, we are doing a pilot RCT in a separate study, where we employ robot-assisted gaming exercise to retrain upper-limb proprioception and motor aspect at one go in stroke rehabilitation. 

163-170 How to ensure randomness and representation of sampling？How to ensure that the distribution of therapists' specialties matches the actual employment data in Singapore？The author mentions that participants are connected with superior therapists, but how to ensure that the participants connected with superior therapists are random and representative.

To the best of our ability, we covered ~10% of the total number of therapists in public service, based on the latest 2019 data released by the Singapore Ministry of Health (that is, 1477 total therapists combined) (Source: https://www.moh.gov.sg/resources-statistics/singapore-health-facts/health-manpower). However, as our survey requires therapists with experience in stroke rehabilitation, and there is no published statistics of therapists specialising only in stroke rehabilitation in Singapore, it is challenging to obtain the precise distribution. 

The supervisors of the therapists would send out a mass email or Whatsapp message to their entire department to respond to the survey without hand picking or selecting which therapist to respond. The therapists who responded are also anonymous to their supervisor and the study team throughout the exercise. 

203-204 Since the author has collected the professional distribution of therapists, why not analyze their respective views on sensory assessment, sensory training, and new technical methods. I think the differences between the specialties of physical therapists and occupational therapists will provide different answers.

We have conducted the additional sets of analyses and the results which are found in the Supporting Information S4 File, Table B, Fig. B, D, F, H, and J. Occupational therapists incorporated technology for retraining upper-limb functions and cognition, whereas physiotherapists paid more attention to lower-limb functions and balance (Line 300-303).

381-383 References are required.

The original statement is more of a possible (hypothetical) reasoning. We have added references (Line 389-392): “Existing interventional studies that target somatosensory loss are still rather limited. One potential reason is the greater emphasis on movement-related interventions to regain functional independence, which are more widely studied and have observable benefits [17, 39, 40]”.

396 I doubt that thermal sense is important for motion control rather than proprioception? Maybe there are other reasons to explain it.

Thank you for clarifying. Thermal sense is not directly critical to voluntary movement, but more for safety during movement (for example, while grabbing a hot cup). We have revised the sentence accordingly (Line 406).

435 The most commonly used assessment methods should be placed first and the least commonly used should be placed later.

As reviewer suggested, we have rearranged the paragraph such that the most commonly used assessment method is placed first (Line 435-446).

---

## [Editor Report · Decision Letter 1]

16 Jun 2022

Current Clinical Practices in Managing Somatosensory Impairments and the Use of Technology in Stroke Rehabilitation

PONE-D-22-10125R1

Dear Dr. Sidarta,

We’re pleased to inform you that your manuscript has been judged scientifically suitable for publication and will be formally accepted for publication once it meets all outstanding technical requirements.

Kind regards,

Benjamin A. Philip

Academic Editor

PLOS ONE
---

## [Editor Report · Acceptance letter]

3 Aug 2022

PONE-D-22-10125R1 

Current Clinical Practice in Managing Somatosensory Impairments and the Use of Technology in Stroke Rehabilitation 

Dear Dr. Sidarta:

I'm pleased to inform you that your manuscript has been deemed suitable for publication in PLOS ONE. Congratulations! Your manuscript is now with our production department. 

Kind regards, 

on behalf of

Dr. Benjamin A. Philip 

Academic Editor

PLOS ONE